# How Epiphytic Are Filmy Ferns? A Semi-Quantitative Approach

Gerhard Zotz [1,2,*] and Helena J. R. Einzmann [1]

1   Institute for Biology and Environmental Sciences, Functional Ecology, Carl von Ossietzky University Oldenburg, P.O. Box 2503, D-26111 Oldenburg, Germany
2   Smithsonian Tropical Research Institute, Balboa, Ancon, Panama City 0843-03092, Panama
*   Correspondence: gerhard.zotz@uol.de; Tel.: +49-(0)441-798-3436

**Abstract:** Similar to plants in many other families, members of the Hymenophyllaceae use numerous substrates for growth, e.g., soil, rocks or tree bark. However, substrate preference does not only differ among species but can also vary among members of the same species. There have been several attempts in the past to appropriately capture this variation, but none proved feasible or was replicated in any subsequent work. In our approach, we use textual information from numerous sources like checklists, floras and species descriptions to come up with a quantitative index of the preference of 450 species of filmy ferns (=c. 75% of all species of the family) for epiphytic, lithophytic or terrestrial growth. We show that the majority of species have clear habitat preferences, while strict habitat specificity is rather uncommon. Our compilation will be an important input for future ecological and phylogenetic studies in this family, but the presented approach is of much more general interest: it is immediately applicable to other taxonomic groups and should eventually allow us to replace the current approach of assigning species to distinct categories (epiphyte, lithophyte or terrestrial) by one that finally reflects biological variability more appropriately.

**Keywords:** accidental epiphyte; epiphyte; facilitation; facultative epiphyte; hemiepiphyte; holoepiphyte; life form; lithophyte





## 1. Introduction

An epiphyte is commonly defined as a non-parasitic plant that grows on another plant throughout its life, without contact to the ground [1]. It has been acknowledged since early on, e.g., [2] that it is impossible to show that this is really true for *all* individuals of a given species; hence, there is some inevitable ambiguity in the terms "epiphytic species" vs. "terrestrial species". Benzing [3] proposed a classification system that distinguished 'true epiphytes', or 'holoepiphytes', from 'facultative epiphytes' and 'accidental epiphytes'. According to this system, epiphytic species are species that (almost) always grow epiphytically and possess specialized characteristics enabling these species to thrive in tree canopies. 'Facultative epiphytes', in turn, are species that occur regularly both on the ground and in the canopy, using sites where terrestrial and arboreal conditions converge, and 'accidental epiphytes' are those that only rarely grow as epiphytes, without any conspicuous modifications related to canopy life.

As an elaboration of this scheme, Ibisch [4] proposed a more quantitative variant in which true epiphytes are defined by the fact that >95% of all members of a species grow epiphytically, with a corresponding proportion of 5–95% for facultative epiphytes and <5% for accidental epiphytes. Later, Burns [5] developed another quantitative approach to classify epiphytes. Although seemingly clear, both approaches face prominent practical problems. First, there were hardly any easily accessible field observations in the literature that could be used. Second, as pointed out by Hoeber and Zotz [6], the spatial scale at which terrestrial and epiphytic occurrence are compared may define the result. As a case in point, they described the observation of a large epiphytic individual of the ornithochorous tree species *Sorbus aucuparia* growing without any terrestrial conspecifics within a forest

area of 1 ha in the Harz mountains in Germany. In this country with a well-studied flora, there is no doubt that this is an accidental epiphytic occurrence of a terrestrial species. However, in areas with more limited botanical exploration, such an observation could be the basis of a publication of an "epiphytic species". This is what happened in the case of *Podocarpus epiphyticus*, originally described based on an epiphytic herbarium specimen in Myanmar [7] but later corrected as an epiphytic individual of the terrestrial species *P. teysmannii* [8].

Yet another problem when trying to quantify the degree of epiphytism of a given species is regional variation. For instance, ferns of the *Polypodium vulgare* complex are usually terrestrial but may occur as obligate epiphytes with high arboreal abundances in certain regions, e.g., [9–11]. Similarly, the bromeliad *Aechmea distichantha* occurs almost entirely epiphytically at the moist end of its distribution but terrestrially or lithophytically at the dry end [12].

Biological variation is often relatively continuous and the outlined problem with the term "epiphyte" is clearly not idiosyncratic to this group of plants. "Halophytes" [13], "CAM plants" [14] or "succulents" are just three of many other cases of similar nature: "Succulence is not a binary trait" [15]. Any study attempting to produce a list of species belonging to such a category has to acknowledge this ambiguity. We have recently compiled a global list of vascular epiphyte species [16]. For each species, at least one reference was given to justify its inclusion. The large majority of the more than 31,000 species are clearly in the category "holoepiphytes". There may be the rare occurrence of an individual plant on other substrates than a tree or a shrub, e.g., a rock, soil or substrates provided by humans such as roofs or power lines [17], but the majority of these plants are (almost) entirely restricted to the epiphytic habitat. There are, however, some taxonomic groups, in which many species seem to be more flexible in their propensity to grow as an epiphyte. For example, of the species categorized as "epiphytes" in the Catálogo de las Plantas Vasculares de Panamá [18] in some families a considerable proportion of species was also listed as terrestrials: Araceae (22%), Gesneriaceae (20%), Ericaceae (25%) or Hymenophyllaceae (23%).

This study focuses on one of these families, Hymenophyllaceae, for which Zotz, Weigelt, Kessler, Kreft and Taylor [16] listed 433 epiphytic species globally, i.e., 72% of an estimated total of c. 600 species. Intraspecific variation was discussed in their study but not addressed in the actual published species list (EpiList 1.0). Some previous treatises with filmy ferns went further in highlighting variation in regard to life form. For example, Ebihara et al. [19] gave detailed life form information for each genus and subgenus of the family, Dubuisson et al. [20] did the same for 193 species of the genus *Trichomanes* s.l. and Lehnert and Krug [21] compiled information for a similar number of species (167) but covered the entire family. However, none of these studies treated the propensity of epiphytic, lithophytic or terrestrial growth as a continuous variable with a truly numeric approach. We did this in the current study, in which we collated life form information for 450 species from numerous sources.

## 2. Materials and Methods

We scanned the taxononomic literature (species descriptions, checklists, floras), the ecological literature (community studies, vegetation descriptions) and numerous web sites and online data bases, e.g., [22] for information on the life form of filmy ferns. Specifically, we collected all available information on the growing sites (tree, rock, soil) of as many species of filmy ferns as possible. However, only sources that did acknowledge that a species may occur on different substrates were considered. Examples of sources that were checked but excluded are Freitas et al. [23], Carvajal-Hernández et al. [24] or Hennequin et al. [25]. In all these papers species were only assigned to a single category. In contrast, e.g., Werner et al. [26] studied epiphytes on remnant trees in Southern Ecuador. Although all species of filmy fern were listed as "epiphytic", we accepted this information

as valid because these authors clearly did distinguish obligate, facultative and accidental epiphytes in their study.

We note that one may miss valid information on species that never occur on the ground with this conservative approach, but including species as 100% epiphytic would inevitably inflate our estimate of "epiphytes" in cases where authors simply found observations of occasional terrestrial occurrence of epiphytes or occasional epiphytic occurrence of terrestrials not worth mentioning. In the end, we could use data from a total of 181 sources (Table S1). Most sources that fulfilled our requirement were annotated checklists and floras. All species names were standardized against Hassler [27] but the original names of the publications can still be found in Table S1.

The translation of literal descriptions of epiphytic, lithophytic or terrestrial occurrences of a given species in proportions is detailed in Table 1.

**Table 1.** Numerical translation of literal descriptions of species occurrences in the original sources as epiphytes, lithophytes or terrestrials. Special cases: with explicit information on life form of specimens we used the proportions, e.g., [28]; "on rotten logs" or "dead wood" counts as "terrestrial" *Trichomanes accedens* in [29], while "epiphytic and on fallen logs" counts as "epiphytic" assuming plants were already on the tree before the fall. We acknowledge that this may obscure possible preferences for rotting logs as growing sites, e.g., [30,31].

| Key to Quantify Verbal Information | % Epiphytic Occurrence (=EV) |
|---|---|
| epiphyte *and* lithophyte (50% to L)/epiphyte *or* terrestrial/facultative epiphyte (50% to T) | 50 |
| epiphyte, lithophyte, or terrestrial (33.3% to L and T) | 33.4 |
| holoepiphyte/hemiepiphyte/epiphyte | 100 |
| obligatory epiphyte | 100 |
| rarely/seldom | 5 |
| very rarely/accidental epiphyte/exceptionally | 1 |
| also as/sometimes/occasionally/uncommonly | 15 |
| commonly/generally/ mainly/mostly/preferentially/primarily/usually | 85 |
| often/frequently/most frequently | 67 |
| less frequently | 33 |

This procedure produced three values per entry, an Epiphyte Value (EV), a Lithophyte Value (LV) and a Terrestrial Value (TV), which add up to unity (100%). While straightforward in those cases in which only occurrences in one of the three habitats were mentioned, it is a major issue to translate vague descriptions like "occasionally epiphytic" or "rarely terrestrial" into a numeric system. This translation is unavoidably arbitrary, but the approach allows a coarse ranking of preferences for epiphytic vs. non-epiphytic growth and is certainly preferable to the usual assignment to just one of several categories, e.g., [16]. Moreover, the consistent use of the rules outlined in Table 1 allows a rapid adjustment of the absolute numbers if there were objective arguments against the used scheme in the future.

Whenever there was information for a given species available in several sources, all sources were considered as equivalent and used without weighting to calculate a single species average of EV, LV and TV (Table S2). Such redundancy should improve the quality of the final estimate, although it is unclear whether different sources really report independent observations or whether life form information may have been copied from another source. The list compiled in this study encompasses 450 species (c. 75% of the family), with an average number of 4.7 sources with occurrence information per species (range 1–45).

For the biological reasons given in the introduction, any number must be considered as a rough estimate. In addition, there are other issues that should be kept in mind when interpreting any single number reported in this study. For example, an entry of 50% epiphyte/50% terrestrial for a species in the final results (Table S2) may have a number of reasons. For one, there may be a single source available that describes the species as

"growing epiphytically and terrestrially". Alternatively, there may be two or more studies, half of which describe the species as "growing epiphytically", the others as "terrestrial" or many alternative combinations. This could reflect an error in some of these sources or rather be based on biological diversity when sources describe varying occurrence patterns in different regions, as described for *Aechmea distichantha* [12]. Since we document all entries in the Supplementary Materials, divergent information about occurrences of a given species can be used as a starting point to study the reasons behind such differences.

We performed two types of numerical analyses. First, we produced a histogram of epiphytic (EV) vs. non-epiphytic (LV + TV) occurrences and compared this distribution to a uniform distribution with a $Chi^2$-test. In a second type of analysis, we followed the approach used by Franco and Silvertown [32] to ordinate demographic components in a triangular space. Similar to the elasticity values from matrix analyses in population biology, the Epiphyte Value, Lithophyte Value and the Terrestrial Value of each species sum to unity, which makes it possible to explore the distribution of filmy ferns in the epiphyte-lithophyte-terrestrial space. For these ordinations, which were done both for the family as a whole and for selected genera, we used the plotrix library version 3.8-1 [33] in R version 4.2.0 [34].

## 3. Results

In a first analysis of the final data set including 2134 entries of 450 species, we lumped terrestrials and lithophytes as non-epiphytes and quantified the degree of epiphytism of each species. If the propensity to grow as an epiphyte were a continuous trait, we would expect the bins of the histogram not to differ much in frequency, while the existence of strong preferences for growth on trees, or not, would lead to a pronounced bimodal distribution with peaks at the extremes. "Obligate" epiphytes (EV > 0.9) clearly make up the single largest bin (accounting for 30% of all species), followed by terrestrial/lithophytic species (EV < 0.1) with some 14% of the total (Figure 1), the distribution deviating significantly from uniformity ($\chi^2 = 261.8$, df = 9, $p < 0.001$). Whether facultative epiphytes cover the whole range from rarely to mostly epiphytic more or less evenly, or whether there may be a third mode of facultative epiphytes with about even probability to occur epiphytically or not is doubtful. Two thirds of the species that fell into the category of even probability to occur epiphytically or not were single entry cases, and thus the existence of a third peak is not strongly supported.

In a second analysis, we used a triangular ordination to include preferences for epiphytic, lithophytic and terrestrial growth in a single analysis (Figure 2). While obligate epiphytes (EV > 0.9) make up just about 30% of all species of filmy ferns, there is clearly a *preference* for epiphytic growth in Hymenophyllaceae. About 68% of all taxa grow primarily as epiphytes, i.e., can be found in the lower left triangle of the epiphyte-lithophyte-terrestrial space, i.e., EV ≥ 0.5, whereas species with LV and TV ≥ 0.5 made up about 11% and 16%, respectively (note that in these calculations we assigned half of the species at the border, e.g., EV = LV = 50% to either E or L). Obligate lithophytes (LV > 0.9) and obligate terrestrial taxa (TV > 0.9) make up less than, respectively, 4% and 6% of all species. A similarly small number of species could be called "generalists" with <50% preference for any of the three growing sites (6%).

A third analysis explored the distribution of four species-rich genera of the Hymenophyllaceae in the epiphyte-lithophyte-terrestrial space (Figure 3). The different genera differ strongly in their habitat preferences, but invariably the majority of species are not restricted to one habitat. On average, however, *Abrodictyum* species show a clear preference for terrestrial growth, while *Didymoglossum* and *Hymenophyllum* species use mostly epiphytic growing sites and, to a lesser extent, rocks, while preferences are most evenly distributed among *Trichomanes* species.

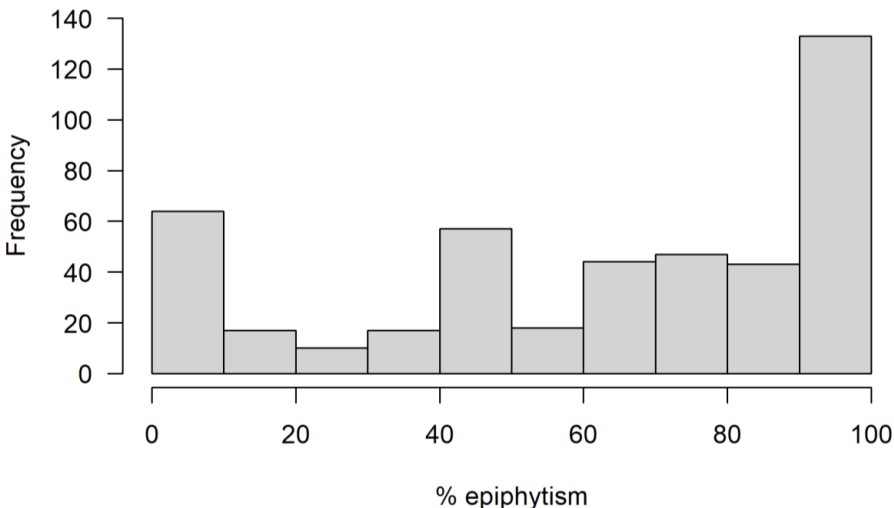

**Figure 1.** Histogram showing the tendency of 450 species of Hymenophyllaceae to occur epiphytically.

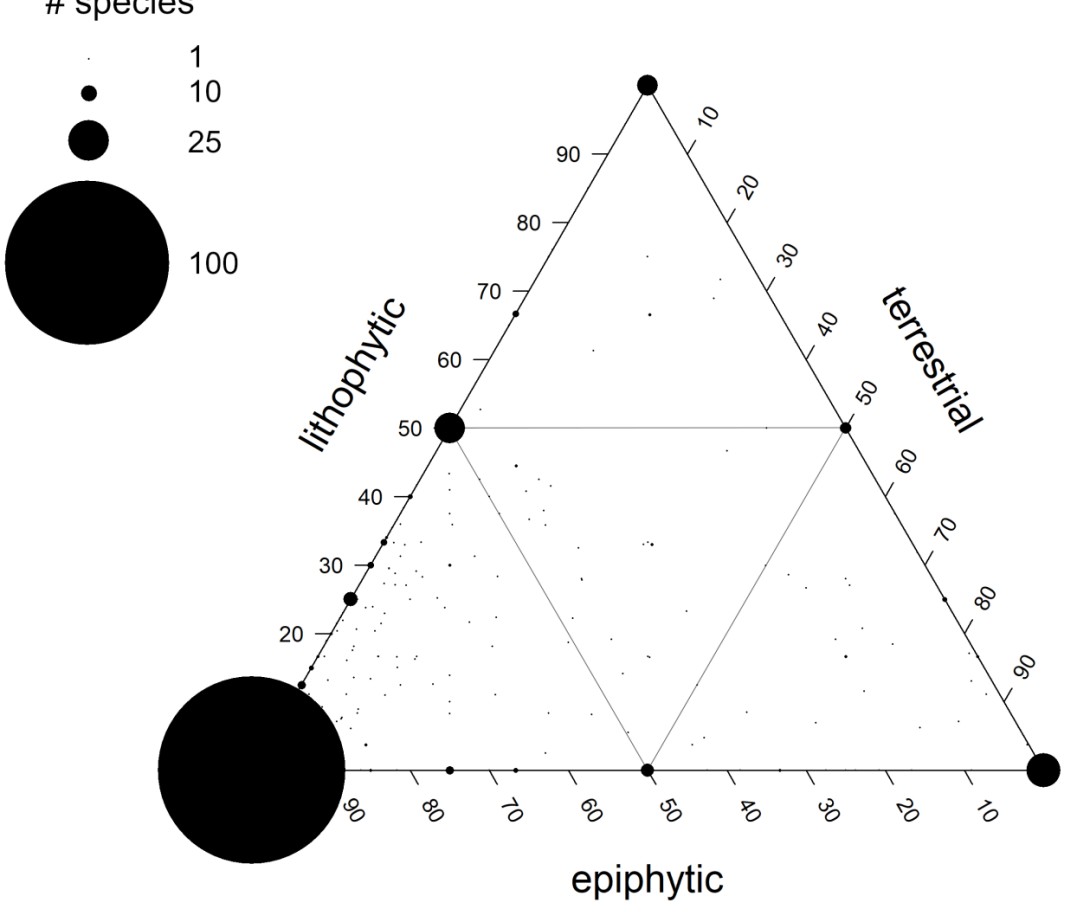

**Figure 2.** Distribution of 450 species of filmy ferns including members of all nine accepted genera in the epiphyte-lithophyte-terrestrial space. The number of species with a preference for epiphytic growth, i.e., E ≥ 50, was 304 (note that in this calculation we assigned half of the species at the border, e.g., EV = LV = 50% to E and L). Symbol size varies with the number of species that share the same values. Individual species values are based on 1–45 sources. The grey lines separate three zones of preference for epiphytic, lithophytic or terrestrial growth, with the central triangle being occupied by generalists. The full data set is given in Tables S1 and S2.

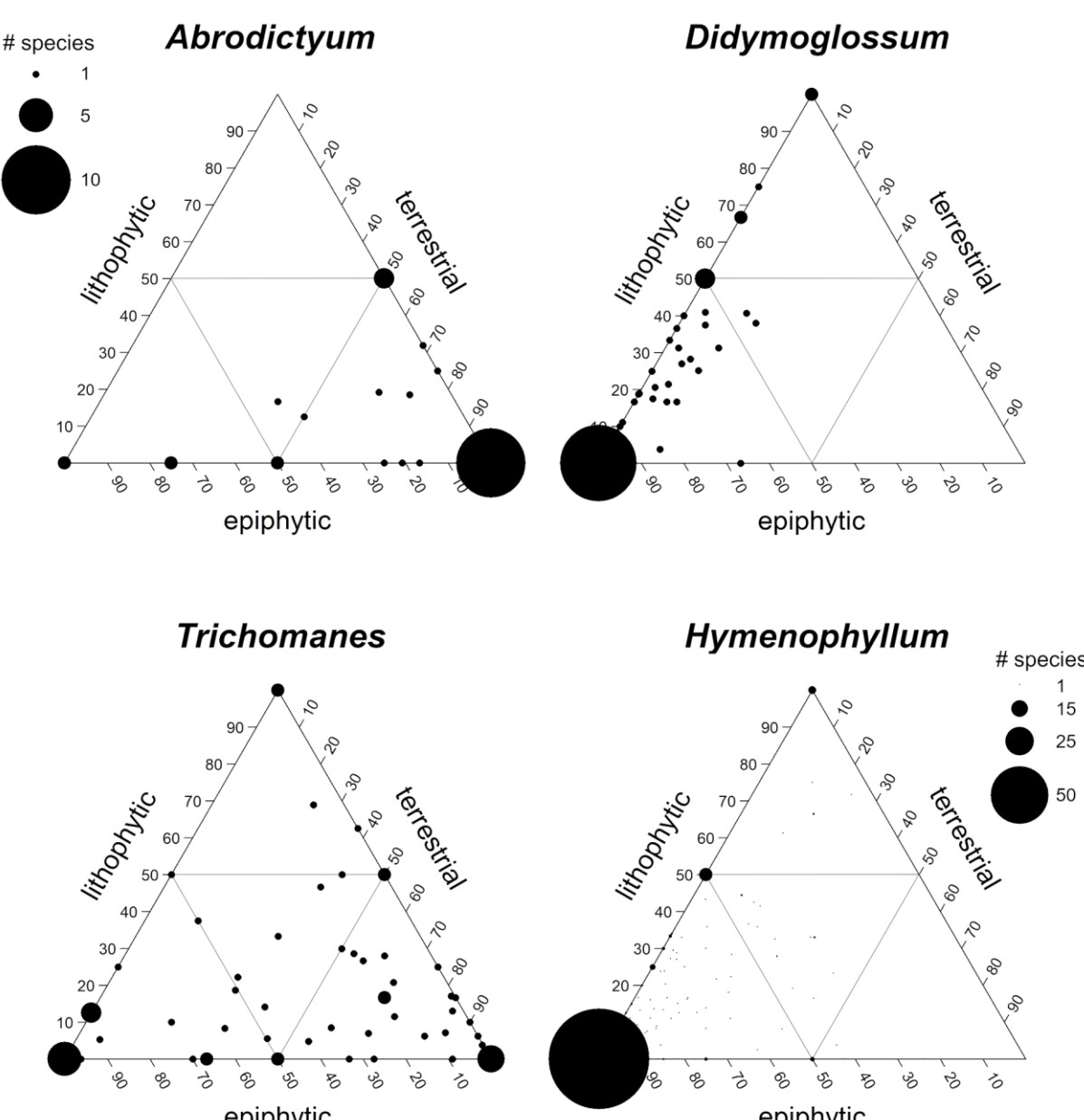

**Figure 3.** Distribution of 28 *Abrodictyum*, 47 *Didymoglossum*, 61 *Trichomanes* and 233 *Hymenophyllum* species in the epiphyte-lithophyte-terrestrial space. Symbol size varies with the number of species with the same values of EV, LV and TV. Scaling is identical for *Abrodictyum*, *Didymoglossum* and *Trichomanes* but differs for *Hymenophyllum*.

## 4. Discussion

Both histogram (Figure 1) and the two ordinations (Figures 2 and 3) are highly informative visualisations of the preference of the family Hymenophyllaceae as a whole or individual genera for epiphytic, lithophytic or terrestrial growth, which goes far beyond the typical categorisation as either epiphyte or terrestrial/epiphyte [16,35] or the verbal descriptions in, e.g., Ebihara, Dubuisson, Iwatsuk, Hennequin and Ito [19]. While each individual data point in our data set is certainly questionable, the overall patterns are probably quite robust. Most importantly, by collating and presenting these data, we expose them to further scrutiny. Using the figures together with Tables S1 and S2, it is possible to critically analyse the intraspecific variation in habitat preference of hundreds of species.

This should either lead to the correction of erroneous descriptions in the literature or allow the interesting study of the nature of the real biological variation.

*How good is the match of the current analysis, EpiList 1.0 and previous descriptions of individual genera?*

Typically, the tendency for epiphytism in a particular genus or family has been assessed by the relative number of "epiphytic species", e.g., [16,35] or verbal characterisations, e.g., [19]. The current study introduces a new method. The results of all three approaches, which are summarized in Table 2, are generally numerically consistent, particularly in the more species-rich genera.

**Table 2.** Average values of the Epiphyte value (EV), lithophyte value (LV) and terrestrial value (TV) in the current analysis, the proportion of all species in the nine accepted genera or the family as a whole with an EV $\geq$ 0.5 and the proportion of epiphyte species per genus or entire family in EpiList 1.0 [16].

| Genus | Description in [19] | EV | LV | TV | EV $\geq$ 0.5 | % Epiphytes in EpiList 1.0 |
|---|---|---|---|---|---|---|
| *Abrodictyum* | Terrestrial in humid places, or epiphytic on tree trunks | 19 | 14 | 67 | 21 | 37 |
| *Callistopteris* | Usually epilithic, occasionally epiphytic | 1 | 8 | 91 | 0 | 40 |
| *Cephalomanes* | Terrestrial or epilithic | 5 | 15 | 80 | 0 | 20 |
| *Crepidomanes* | Mostly epiphytic | 64 | 32 | 4 | 80 | 71 |
| *Didymoglossum* | Epilithic or low-epiphytic | 71 | 24 | 5 | 85 | 92 |
| *Hymenophyllum* | Usually low- to middle-epiphytes on tree trunks, sometimes canopy sun epiphytes, occasionally epilithic or terrestrial | 75 | 18 | 7 | 85 | 89 |
| *Polyphlebium* | Epilithic or epiphytic | 78 | 15 | 7 | 85 | 88 |
| *Trichomanes* | Usually terrestrial in humid places, sometimes epiphytic on tree trunks | 36 | 15 | 49 | 34 | 57 |
| *Vandenboschia* | Hemi-epiphytic on tree trunks or epilithic, occasionally terrestrial | 64 | 24 | 12 | 74 | 92 |
| **whole family** | | *63* | *19* | *18* | *71* | *72* |

For example, most *Abrodictyum* species prefer a terrestrial habitat as described by Ebihara, Dubuisson, Iwatsuk, Hennequin and Ito [19]; the genus has also the highest TV and the lowest proportion of epiphyte species in EpiList 1.0, the opposite being true for members of the genus *Didymoglossum*. However, a strict preference of a *Didymoglossum* species for growth on a tree or on a rock is relatively rare: for the majority of species, growth on more than one substrate has been reported. Assuming that rare occurrences of, e.g., a typically epiphytic species on a rock or epiphytic occurrence of a typical terrestrial or lithophytic species probably remain unreported, strict preference may even be less frequent than reflected in our data set. Noteworthy, the number of "epiphyte species" in EpiList 1.0 is always higher than the average EV of the studied genera or the family as a whole (Table 2). This consistent difference is easily explained by the dissimilar approaches in the two studies. In the former [16], species were included when at least one reference in the scientific literature described that species as epiphytic or primarily epiphytic, while in the current study we actually quantified habitat preference using as many sources as possible. For example, *Trichomanes egleri* and *T. pinnatum* were described as facultative epiphytes in Hokche et al. [36] and Steyermark et al. [37], respectively, which led to their inclusion in EpiList 1.0, but the current analysis based on numerous sources suggests that epiphytic occurrences are too rare overall as to justify their inclusion. However, any discrepancy depends on the definition of "epiphyte species". Confronted with a similar problem in the case of plants that show some degree of nocturnal acidification, Winter [38] defined "CAM species" as those that obtain the majority of their carbon through the CAM pathway throughout their lives, typically deduced from a $\delta^{13}C$ values of leaf tissue $> -20$ ‰. Using a similar rationale, we may define "epiphyte species" as those taxa with an EV $\geq$ 0.5, i.e., species that occur *primarily* in tree crowns. In Figures 2 and 3,

this would be equivalent to all species in the lower left of the four triangles within the epiphyte-lithophyte-terrestrial space. For the family as whole, this would yield 71%, which is remarkably close to the 72% of epiphyte "species" in EpiList 1.0 (Table 2). For individual genera, the outcome of such a comparison is more diverse, with very similar numbers in *Hymenophyllum* and *Didymoglossum* but quite different outcomes in smaller genera like *Callistopteris* or *Cephalomanes* (Table 2).

*Caveats*

The quality of an analysis as the present one depends on the quality of the sources. A major problem is the inconsistent use of terminology, even within a single paper. For example, Ebihara, Dubuisson, Iwatsuk, Hennequin and Ito [19] state that "all species in *Didymoglossum* are dwarf epiphytes, mainly in tropical regions", only to specify a few lines later that their habitat is "epilithic or low-epiphytic". Similarly, *Polyphlebium endlicherianum* is called "epiphyte" in the Flora of the Marquesas Islands [39], but the habitat is described as "trees and rocks". In such cases, we used the habitat details to score a species. Apart from such inconsistencies, there is a tendency among many researchers to prefer neat categories over "noise". If explicitly mentioned, one can at least reject such data in an analysis like the current one. As an example, Mellado-Mansilla et al. [40], noted that a large proportion of the species described as "epiphytes" by Moreno et al. [41], were actually found primarily on rocks or slopes of river beds at their study site but classified them as epiphytes as the information in Moreno, Le Quesne, Díaz and Rodríguez [41] was taken as "the most common growth forms for Andean temperate forest species". Similarly, Chen et al. [42], used the "most common description" if there was a conflict between published studies, with an aim to avoid scoring taxa as polymorphic. Such approaches produce neat categories but obscure biological variation. In many cases, however, such an approach is not explicitly mentioned. For that reason, we did not include the life form information of numerous studies.

We are also aware that our numerical approach is not without problems. First, we obviously cannot assume that terms like "commonly" or "rarely" are written with the same numerical equivalent in mind in the many sources included in this study. Second, to assign a 50% preference for E and L for "on trees and rocks" was simply the most parsimonious approach for want of a more convincing alternative. Due to the fact that we typically had several entries per species with at least slightly divergent information, there were only 7% of all species with 50%/50% averages for EV, LV and TV (Figure 1). We also realize that life form categorisations in a large flora, e.g., [43], a checklist, e.g., [36], or a large review, e.g., [44] integrate much more information than a report about one particular locality, e.g., [45,46], but we were not sure how exactly to weigh sources and again used the most parsimonious option. In the end, our analysis should simply be taken as semi-quantitative.

*The hemiepiphyte issue*

So-called "hemiepiphytic" species and climbers pose a special problem in a study that distinguishes between epiphytic, lithophytic and terrestrial growth. As discussed in detail in Zotz et al. [47], hemiepiphytic vascular plants are defined by their ontogeny. In the case of hemiepiphytic ferns, gametophytes start epiphytically on a tree and the sporophyte, which is initially epiphytic as well, later establishes root contact with the soil, i.e., the sporophyte would be categorized as terrestrial. Thus, hemiepiphytes do not neatly fall in any of the three basic categories of our study.

Hemiepiphytic growth in ferns has been originally demonstrated for *Vandenboschia collariata* by Nitta and Epps [48] and subsequently for a number of other fern taxa in several families [47,49,50]. Although other *Vandenboschia* or *Trichomanes* species have been given the label hemiepiphyte repeatedly, e.g., [19,29], we are not aware that this has actually been shown unambiguously for these and any other species in the genus or the family. Unfortunately, researchers routinely deduce ontogeny from single observations, which is bound to produce dubious results. Moreover, the labels "hemiepiphyte", "climber", "epiphyte" are very inconsistently used in the different sources, particularly for *Vandenboschia*. However, the genus is relatively species-poor, and thus including/excluding *Vandenboschia*

does hardly change the overall result—we show the ordination without the genus in the Supplementary Materials as Figure S1. Species in other genera, e.g., *Trichomanes tanaicum* (subgenus Lacostea) are also sometimes labelled hemiepiphyte, e.g., [29], again without evidence. A better understanding of the occurrence of hemiepiphytism in this and other families is needed and is particularly important in the context of evolutionary studies. Frequently, hemiepiphytes seem to be intermediates in the transition from terrestrial to epiphytic growth and vice versa [42,51], but such an analysis depends on the correct identification of the life form of the studied species.

*Towards a more realistic view of epiphytes* **vs.** *non-epiphytes*

Clearly, if we want to understand the ecology of Hymenophyllaceae and other ferns and the mechanistic basis of habitat preferences, the correct assessment of their growing sites is an indispensable requisite. Acknowledging variation is also critical for analyses of trait evolution. Many such analyses use simple dichotomies, e.g., [25], but there are others which at least use a "mixed" category [52] or even more elaborate schemes [21]. We argue that treating evolutionary transitions from terrestrial/lithophytic to epiphytic growth [25] or vice versa [42] as a step change instead of a gradual change is probably not an appropriate description of the actual developments. Acknowledging facultative life forms should enrich any such analysis. Similarly, treating, e.g., lithophytic, terrestrial and epiphytic habitats as inherently distinct is also an oversimplification. On the one hand, there can be much overlap in growth conditions; on the other hand, there is much variation within each of these three habitat types, e.g., [53]. For example, the actual growth conditions on moss-covered rocks in the understory or on the lower portions of similarly moss-covered tree trunks may vary little for a fern, but bark and rock do differ in many other aspects, e.g., in thermal properties or in stability [1]. In contrast, the growth conditions of two lithophytes may be quite distinct, one growing in the open may experience rather dry conditions with high temperature peaks, while conditions on a shaded rock near a creek may resemble those of a rheophyte.

The range of different environmental conditions of epiphytic filmy ferns is probably not as large as in other taxa with many epiphytic members, e.g., in bromeliads or orchids, which may be found from the shaded understory to very exposed sites in the uppermost crown of trees. In general, filmy ferns are restricted to more shaded and humid growing sites in the understory, but there are reports of at least occasional growth in higher parts of the forest canopy [54,55].

In summary, we present a new semi-quantitative approach to assessing preferences of filmy ferns for particular growing sites. Consistent with conclusions of Lehnert and Krug [21], we find that a majority of species show clear preferences to either epiphytic, lithophytic or terrestrial growth, while strict specialisation is not as common as may be deduced from species lists such as EpiList 1.0. Our approach should be very useful in phylogenetic and ecological studies in this particular family, but the method we used is of much more general interest. It is straightforward to apply it to other genera or families, which will allow us to replace the current approach of classifying species into distinct categories (epiphytic, lithophytic or terrestrial) with a much more realistic one that reflects the more continuous variation observed in nature.

**Supplementary Materials:** The following supporting information can be downloaded at: https://www.mdpi.com/article/10.3390/d15020270/s1, Figure S1: Distribution of filmy ferns in the epiphyte-lithophyte-terrestrial space excluding *Vandenboschia*; Table S1: Original data from 181 sources; Table S2: Mean growth preference values for 450 species [56–223].

**Author Contributions:** Conceptualization, G.Z.; Methodology, G.Z. and H.J.R.E.; Formal Analysis, G.Z. and H.J.R.E.; Data Curation, G.Z.; Writing—Original Draft Preparation, G.Z.; Writing—Review & Editing, G.Z. and H.J.R.E.; Visualization, G.Z. and H.J.R.E. All authors have read and agreed to the published version of the manuscript.

**Funding:** This research received no external funding.

**Institutional Review Board Statement:** Not applicable.

**Data Availability Statement:** All data are given in the Supplementary Materials.

**Acknowledgments:** Thanks for help with the data input to Norbert Wagner (Oldenburg), Ibrahim Elias (Oldenburg), Sarah Petrasz (Oldenburg).

**Conflicts of Interest:** The authors declare no conflict of interest.

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
