# Peer review of "How Epiphytic Are Filmy Ferns? A Semi-Quantitative Approach"

_diversity, doi:10.3390/d15020270_

Round 1
Reviewer 1 Report
The article submitted to review deals with two interesting topics, i.e. very specific group of leptosporangiate ferns, so called filmy ferns (Hymenophyllaceae), and the plasticity of ferns regarding the substrate they occupy. The Authors have done a tremendous job collecting data from different studies on over 400 species, as well as synonymizing their names according to the current state. The manuscript contains an interesting review of problems related to the description and classification of epiphytes. Figures 2 and 3 present the results is clearly and simply
However, the presented study raises also some doubts.
The title suggests that the Authors analyzed a group of species representative of the family, but does not include representatives of all or the most genera; only four of nine distinguished genera were taken to analysis. The Authors analyzed 73% of the subfamily Hymenophylloideae containing one and the most numerous genus Hymenophyllum, and the results are representative of this unit. However, only 50% of Trichomanoideae were analyzed. This subfamily is more diverse, contains eight genera and 50% is not enough to draw conclusions for the whole unit. It would be appropriate to specify the title, e.g.[…] A quantitative approach by example of Abrodictyum, Didymoglossum, Trichomanes and Hymenophyllum or […] by example of selected genera.
The Authors report that they have analyzed 75% of the species of Hymenophyllaceae, but the data provided in the study varies. In the text (line 165) and the descriptions of Fig.1 and 2 give 446 species (c. 76%) while the description of the same fig. in appendix gives 427 species (c. 72%). Which value is correct?
In the discussion, the authors pointed out that the sources on which they based their analysis provided imprecise and sometimes even contradictory data. In this situation, the results of analysis are uncertain. This is not the fault tor mistake of the Authors, however the conclusions drawn on their basis are rather suggestions or even speculation. Perhaps if species were linked to a climatic zone or vegetation formation, and compared to the results of the analysis, data would be credible.
In general, the work is more about testing the method and presenting its deficiency on the example of representatives of the Hymenophyllaceae family than characterizing the family using the presented method. The title should be changed and the conclusions should be given in a more presumed form.
The proposed method is interesting and seems to be helpful in determining and presenting the preferences of substrate or habitat of plants and living organisms.
Necessary corrections
Figure 2 and 3 is # species, should be probably % species
Two forms of notation Epilist and EpiList should be unified.
Appendix Table 1 is sorted alphabetically, however with little disturbances (Hymenophyllum acanthoides/acutum/acanthoides).
Author Response
The article submitted to review deals with two interesting topics, i.e. very specific group of leptosporangiate ferns, so called filmy ferns (Hymenophyllaceae), and the plasticity of ferns regarding the substrate they occupy. The Authors have done a tremendous job collecting data from different studies on over 400 species, as well as synonymizing their names according to the current state. The manuscript contains an interesting review of problems related to the description and classification of epiphytes. Figures 2 and 3 present the results is clearly and simply
Thanks for this positive reception of our paper
However, the presented study raises also some doubts.
The title suggests that the Authors analyzed a group of species representative of the family, but does not include representatives of all or the most genera; only four of nine distinguished genera were taken to analysis. The Authors analyzed 73% of the subfamily Hymenophylloideae containing one and the most numerous genus Hymenophyllum, and the results are representative of this unit. However, only 50% of Trichomanoideae were analyzed. This subfamily is more diverse, contains eight genera and 50% is not enough to draw conclusions for the whole unit. It would be appropriate to specify the title, e.g.[…] A quantitative approach by example of Abrodictyum, Didymoglossum, Trichomanes and Hymenophyllum or […] by example of selected genera.
This is a misunderstanding. Our analysis included large proportions of all nine genera with 60 – 100% of all described species per genus. This can also be seen by looking at the Appendices. But to avoid any confusion we now add an explicit statement in the text and extend the analysis of Table 2 to all genera
The Authors report that they have analyzed 75% of the species of Hymenophyllaceae, but the data provided in the study varies. In the text (line 165) and the descriptions of Fig.1 and 2 give 446 species (c. 76%) while the description of the same fig. in appendix gives 427 species (c. 72%). Which value is correct?
We added new entries and revised all numbers. Coincidentally, the Figure in the Appendix is of course lower because it was constructed without the Vandenboschia species
In the discussion, the authors pointed out that the sources on which they based their analysis provided imprecise and sometimes even contradictory data. In this situation, the results of analysis are uncertain. This is not the fault tor mistake of the Authors, however the conclusions drawn on their basis are rather suggestions or even speculation. Perhaps if species were linked to a climatic zone or vegetation formation, and compared to the results of the analysis, data would be credible.
We make a major effort to highlight the possible problems with the data.
In general, the work is more about testing the method and presenting its deficiency on the example of representatives of the Hymenophyllaceae family than characterizing the family using the presented method. The title should be changed and the conclusions should be given in a more presumed form.
As suggested, we changed the title to “semi-quantitative”. We disagree, however, in regard to the suggestion that this paper is not really about filmy ferns. We assume this notion is cause by the misunderstanding about the inclusion of all genera. Thus, we see no reason to change the conclusions.
The proposed method is interesting and seems to be helpful in determining and presenting the preferences of substrate or habitat of plants and living organisms.
Necessary corrections
Figure 2 and 3 is # species, should be probably % species
No, number of species is correct, because it is the number of species within the ELT-space for any given %EV - %LV combination.
Two forms of notation Epilist and EpiList should be unified.
Changed as suggested
Appendix Table 1 is sorted alphabetically, however with little disturbances (Hymenophyllum acanthoides/acutum/acanthoides).
corrected
Reviewer 2 Report
There are many limitations to this work, as outlined by the authors, but the new perspective it offers is an important contribution to our understanding of habitat preferences in filmy ferns. They analyzed published information from a large number of species (446), and thus, it seems the most comprehensive survey and assessment on the subject thus far. Although the grouping of the samples into 3 main types of habitats (habits?) preferences is an oversimplification, it is a good starting point considering the complicated nature of the data and sources. I commend the authors for their meticulous analysis of a complicated subject. Their results show the relative distribution in space for each of the 3 main types of life-forms (habit) in filmy ferns, analyzed based on family and representative species-rich genera level (although one can dig deeper for species level info from the supplementary material). Although the whole family mostly exhibits epiphytic habit, there are differences in habit types at genus level, i.e., Didymoglossum and Hymenophyllum are mostly epiphytic, while Abrodictyum and Trichomanes are mostly terrestrial. Most importantly, their distribution in space analysis also shows the gradation between one type to the other and thus fills the gaps in our understanding of the variation that exists out there. Such information may also have important implications in phylogenetic and ecological analyses.
Specific comments:
I struggle with the terminologies used in this paper (some were defined but vague and others not defined), but most importantly, the use of the term “epiphytic habitat” is problematic for me. I believe it should be “epiphytic habit”, which focuses on the life-form not the substrate (habitat). Epiphytic growth and growth form are also not appropriate terms in this paper’s context.
The Abstract mentions “mosses” that seems to constitute a 4th type of habitat, but it’s not considered in the rest of the paper as such. It should be deleted from the abstract and the mention of moss-covered rocks and trees in the other parts of the paper as should be sufficient to convey the concept. The term “conspecifics” is mentioned a lot but wasn’t directly addressed in the rest of the paper (other than mentioning it) to warrant being highlighted in the introduction. State 446 instead of “more than 400 species”.
Obviously, the “quantitative” analysis is debatable, but the authors already acknowledged that their paper is more of a semi-quantitative analysis. However, it seems more appropriate to consider this paper as a numerical (translation) analysis, a term they also used a lot in their paper.
In Table 1, it is not clear on how some of the numbers (e.g., 1, 5, etc.) were derived. A brief description in the caption should be added.
Author Response
There are many limitations to this work, as outlined by the authors, but the new perspective it offers is an important contribution to our understanding of habitat preferences in filmy ferns. They analyzed published information from a large number of species (446), and thus, it seems the most comprehensive survey and assessment on the subject thus far. Although the grouping of the samples into 3 main types of habitats (habits?) preferences is an oversimplification, it is a good starting point considering the complicated nature of the data and sources. I commend the authors for their meticulous analysis of a complicated subject. Their results show the relative distribution in space for each of the 3 main types of life-forms (habit) in filmy ferns, analyzed based on family and representative species-rich genera level (although one can dig deeper for species level info from the supplementary material). Although the whole family mostly exhibits epiphytic habit, there are differences in habit types at genus level, i.e., Didymoglossum and Hymenophyllum are mostly epiphytic, while Abrodictyum and Trichomanes are mostly terrestrial. Most importantly, their distribution in space analysis also shows the gradation between one type to the other and thus fills the gaps in our understanding of the variation that exists out there. Such information may also have important implications in phylogenetic and ecological analyses.
Thanks for this positive feedback.
Specific comments:
I struggle with the terminologies used in this paper (some were defined but vague and others not defined), but most importantly, the use of the term “epiphytic habitat” is problematic for me. I believe it should be “epiphytic habit”, which focuses on the life-form not the substrate (habitat). Epiphytic growth and growth form are also not appropriate terms in this paper’s context.
We agree that terminology is not straightforward – while an ”epiphytic individual” is unambiguous, the claim that “species X is an epiphyte” is impossible to prove as it means that 100% of all individuals of this species grow “in the epiphytic habitat”. Therefore we refrain from the usual “species X is an epiphyte” in the entire literature, and actually think that “epiphytic habitat” vs. “terrestrial habitat” is well chosen because it implies that the growth of an individual plant on a tree does not make the species an epiphyte. We make this clear from the start with the first sentences of the introduction. The last sentence leaves us puzzled: 1) “Epiphytic growth” is actually at the heart of this paper: we use it to reflect that an individual plant may grow epiphytically without making the species an “epiphyte”. 2) we do not use “growth form” – the term appears only once as a literal citation
The Abstract mentions “mosses” that seems to constitute a 4th type of habitat, but it’s not considered in the rest of the paper as such. It should be deleted from the abstract and the mention of moss-covered rocks and trees in the other parts of the paper as should be sufficient to convey the concept. The term “conspecifics” is mentioned a lot but wasn’t directly addressed in the rest of the paper (other than mentioning it) to warrant being highlighted in the introduction. State 446 instead of “more than 400 species”.
“Moss” has been deleted. The word “conspecifics” is used exactly twice! It simply means a member of the same species. Since intraspecific variation is the central point of this paper, it is essential to have this mentioned in the abstract, but we have now replaced the word by “members of the same species”. Finally, we have now given the exact number of species.
Obviously, the “quantitative” analysis is debatable, but the authors already acknowledged that their paper is more of a semi-quantitative analysis. However, it seems more appropriate to consider this paper as a numerical (translation) analysis, a term they also used a lot in their paper.
We agree and to make this point from the start - we changed the title
In Table 1, it is not clear on how some of the numbers (e.g., 1, 5, etc.) were derived. A brief description in the caption should be added.
We already discuss this extensively in lines 115 -131. It is simply a reasonable guess what, e.g., “rarely epiphytic” may mean in regard to the relative occurrence of E vs. L and T. We point out that it would be straightforward to use other numbers. However, since we cannot assume that different authors have the same concept of “rare”, this seems rather pointless to us. We also stress in lines 132 – 142 that variation between sources result from such differences or be due to biological variation. With our data set we don’t see how we should tackle this question but hope our analysis stimulates studies that can use our data set as a starting point.